# SimTeG: A Frustratingly Simple Approach Improves Textual Graph Learning

## Abstract

Textual graphs (TGs) are graphs whose nodes correspond to text (sentences or documents), which are widely prevalent. The representation learning of TGs involves two stages: ($i$) *unsupervised feature extraction* and ($ii$) *supervised graph representation learning*. In recent years, extensive efforts have been devoted to the latter stage, where Graph Neural Networks (GNNs) have dominated. However, the former stage for most existing graph benchmarks still relies on traditional feature engineering techniques. This motivates us to investigate the outcomes of enhancing only the text embeddings in benchmark models. While it is anticipated that advanced text embeddings will boost GNN performance, key questions remain underexplored: the extent of this improvement, particularly how advanced text features can enhance a rudimentary GNN architecture. Therefore, in this work, we present SimTeG, a frustratingly Simple approach for Textual Graph learning that does not innovate in frameworks, models, and tasks. We first perform *supervised* parameter-efficient fine-tuning (PEFT) on a pre-trained LM on the downstream task, such as node classification. We then generate node embeddings using the last hidden states of finetuned LM. These derived features can be further utilized by any GNN for training on the same task. We evaluate our approach on two fundamental graph representation learning tasks: *node classification* and *link prediction*. Through extensive experiments, we show that our approach significantly improves the performance of various GNNs, *especially basic GNN baselines*, on multiple graph benchmarks. Remarkably, when additional supporting text provided by large language models (LLMs) is included, a simple two-layer GraphSAGE trained on an ensemble of SimTeG achieves an accuracy of 77.48% on `OGBN-Arxiv`, comparable to state-of-the-art (SOTA) performance obtained from far more complicated GNN architectures. We will release our code and generated node features soon.

## 1 Introduction

Textual Graphs (TGs) offer a graph-based representation of text data where relationships between phrases, sentences, or documents are depicted through edges. TGs are ubiquitous in real-world applications, including citation graphs (Hu et al., 2020; Yang et al., 2016), knowledge graphs (Wang et al., 2021), and social networks (Zeng et al., 2019; Hamilton et al., 2017), provided that each entity can be represented as text. Different from traditional NLP tasks, instances in TGs are correlated with each other, which provides non-trivial and specific information for downstream tasks. In general, graph benchmarks are usually task-specific (Hu et al., 2020), and most TGs are designed for two fundamental tasks: *node classification* and *link prediction*. For the first one, we aim to predict the category of unlabeled nodes while for the second one, our goal is to predict missing links among nodes. For both tasks, text attributes offer critical information.

In recent years, TG representation learning follows a two-stage paradigm: ($i$) *upstream: unsupervised feature extraction* that encodes text into numeric embeddings, and ($ii$) *downstream: supervised graph representation learning* that further transform the embeddings utilizing the graph structure. While Graph Neural Networks (GNNs) have dominated the latter stage, with an extensive body of academic research published, the former stage surprisingly still relies on traditional feature engineering techniques. For example, in most existing graph benchmarks (Hu et al., 2020; Yang et al., 2016; Zeng et al., 2019), node features are constructed using skip-gram (Mikolov et al., 2013). This

intuitively limits the performance of downstream GNNs, as it fails to fully capture textual semantics, fostering an increasing number of GNN models with more and more complex structures.

Consequently, our research aims to investigate the impact of enhancing benchmark text embeddings exclusively. While an anticipated outcome is an improved GNN performance by introducing advanced text embeddings, key inquiries remain underexplored: the extent of this improvement, and specifically, the potential enhancement of a basic GNN architecture by advanced text features. This inquiry holds practical significance, as industry applications of GNN architectures are limited by computational efficiency. To date, a notable exception is Pinsage (Ying et al., 2018), a GraphSAGE-based recommendation system for Pinterest. If incorporating advanced text features could bypass the necessity of using complex GNN models, it would significantly boost the application of GNNs in industry. We take an step forwards to explore the above research questions by introducing a simple and straightforward framework SimTeG on TGs and empirically evaluating it on two fundamental graph tasks: node classification and link prediction. We first parameter-efficiently finetune (PEFT) an LM on the textual corpus of a TG with task-specific labels and then use the finetuned LM to generate node representations given its text by removing the head layer. Afterward, a GNN is trained with the derived node embeddings on the *same* downstream task for final evaluation. with extensive experiments on three prestigious graph benchmarks on node classification and link prediction, we find several *key observations*:

❶ Good language modeling could generally improve the learning of GNNs on both node classification and link prediction. We evaluate SimTeG on three prestigious graph benchmarks for either node classification or link prediction, and find that SimTeG consistently outperforms the official features and the features generated by pretrained LMs (without finetuning) by a large margin. Notably, backed with SOTA GNN, we achieve *new SOTA performance* of $78.02\%$ on `OGBN-Arxiv`. See Sec. 5.1 and Appendix A1 for details.

❷ Incorporating advanced text features, a simple two-large GraphSAGE achieves on-par SOTA performance on node classfication and link prediction tasks. Notably, a simple two-layer Graph-SAGE (Hamilton et al., 2017) trained on SimTeG with proper LM backbones achieves on-par SOTA performance of $77.48\%$ on `OGBN-Arxiv` (Hu et al., 2020). To date, It achieves the top three rank on the leaderboard, while the original result for sole GraphSAGE is ranked 62.

❸ PEFT are crucial when finetuning LMs to generate representative embeddings, because full-finetuning usually leads to extreme overfitting due to its large parameter space and the caused fitting ability. The overfitting in the LM finetuning stage will hinder the training of downstream GNNs with a collapsed feature space. See Sec. 5.3 for details.

❹ SimTeG is moderately sensitive to the selection of LMs. Generally, the performance of SimTeG is positively correlated with the corresponding LM's performance on text embedding tasks, e.g. classification and retrieval. In addition, the performance is not closely correlated with the number of parameters in the LM. We refer to Sec. 5.4 for details. Based on this, we expect further improvement of SimTeG once more powerful LMs for text embedding are available.

## 2    RELATED WORKS

In this section, we first present several works that are closely related to ours and further clarify several concepts and research lines that are plausibly related to ours in terms of similar terminology.

**Leveraging LMs on TGs.** Focusing on leveraging the power of LMs to TGs, there are several works that are existed and directly comparable with ours. For these works, they either focus on $(i)$ *designing specific strategies to generate node embeddings using LMs* (He et al., 2023; Chien et al., 2021) or $(ii)$ *jointly training LMs and GNNs within a framework* (Zhao et al., 2022; Mavromatis et al., 2023). Representatively, for the former one, Chien et al. (2021) proposed a self-supervised graph learning task integrating XR-Transformers (Zhang et al., 2021b) to extract node representation, which shows superior performance on multiple graph benchmarks, validating the necessity for acquiring high-quality node features for attributed graphs. Jin et al. (2023) proposed two pretraining strategies for network-contextualized masked language modeling and masked node prediction to capture semantics and structure information at once. Besides, He et al. (2023) utilizes ChatGPT (OpenAI, 2023) to generate additional supporting text with LLMs. For the latter mechanism, Zhao et al. (2022) proposed a variational expectation maximization joint-training framework for LMs and GNNs to learn powerful graph representations. Mavromatis et al. (2023) designs a graph structure-aware framework

to distill the knowledge from GNNs to LMs. Generally, the joint-training framework requires specific communication between LMs and GNNs, e.g. pseudo labels (Zhao et al., 2022) or hidden states (Mavromatis et al., 2023). It is worth noting that the concurrent work He et al. (2023) proposed a close method to ours. However, He et al. (2023) focuses on generating additional informative texts for nodes with LLMs, which is specifically for citation networks on node classification task. In contrast, we focus on generally investigating the effectiveness of our proposed method, which could be widely applied to unlimited datasets and tasks. Utilizing the additional text provided by He et al. (2023), we further show that our method could achieve now SOTA on `OGBN-Arxiv`. In addition to the main streams, there are related works trying to fuse the architecture of LM and GNN for end-to-end training. Yang et al. (2021) proposed a nested architecture by injecting GNN layers into LM layers. However, due to the natural incompatibleness regarding training batch size, this architecture only allows 1-hop message passing, which significantly reduce the learning capability of GNNs.

**More "Related" Works.** ❶ *Graph Transformers* (Wu et al., 2021; Ying et al., 2021; Hussain et al., 2022; Park et al., 2022; Chen et al., 2022): Nowadays, Graph Transformers are mostly used to denote Transformer-based architectures that embed both topological structure and node features. Different from our work, these models focus on graph-level problems (e.g. graph classification and graph generation) and specific domains (e.g. molecular datasets and protein association networks), which cannot be adopted on TGs. ❷ *Leveraging GNNs on Texts* (Zhu et al., 2021; Huang et al., 2019; Zhang et al., 2020): Another seemingly related line on integrating GNNs and LMs is conversely applying GNNs to textual documents. Different from TGs, GNNs here do not rely on ground-truth graph structures but the self-constructed or synthetic ones.

## 3 PRELIMINARIES

**Notations.** To make notations consistent, we use **bold** uppercase letters to denote matrices and vectors, and calligraphic font types (e.g. $\mathcal{T}$) to denote sets. We denote a textual graph as a set $\mathcal{G} = (\mathcal{V}, \mathcal{E}, \mathcal{T})$, where $\mathcal{V}$ and $\mathcal{E}$ are a set of nodes and edges, respectively. $\mathcal{T}$ is a set of text and each textual item is aligned with a node $v \in \mathcal{V}$. For practical usage, we usually rewrite $\mathcal{E}$ into $\mathbf{A} \in \{0, 1\}^{|\mathcal{V}| \times |\mathcal{V}|}$, which is a sparse matrix, where entry $\mathbf{A}_{i,j}$ denotes the link between node $v_i, v_j \in \mathcal{V}$.

**Problem Formulations.** We focus on two fundamental tasks in TGs: $(i)$ *node classification* and $(ii)$ *link prediction*. For node classification, given a TG $\mathcal{G}$, we aim to learn a model $\Phi : \mathcal{V} \to \mathcal{Y}$, where $\mathcal{Y}$ is the ground truth labels. For link prediction, given a TG $\mathcal{G}$, we aim to learn a model $\Phi : \mathcal{V} \times \mathcal{V} \to \{0, 1\}$, where $f(v_i, v_j) = 1$ if there is a link between $v_i$ and $v_j$, otherwise $f(v_i, v_j) = 0$. Different from traditional tasks that are widely explored by the graph learning community, evolving original text into learning is non-trivial. Particularly, when ablating the graphs structure, node classification and link prediction problem are collapsed to text classification and text similarity problem, respectively. This sheds light on how to leverage LMs for TG representation learning.

**Node-level Graph Neural Networks.** Nowadays, GNNs have dominated graph-related tasks. Here we focus on GNN models working on node-level tasks (i.e. *node classification* and *link prediction*). These models work on generating node representations by recursively aggregating features from their multi-hop neighbors, which is usually noted as *message passing*. Generally, one can formulate a graph convolution layer as: $\boldsymbol{X}_{l+1} = \Psi_l(\boldsymbol{C}\boldsymbol{X}_l)$, where $\boldsymbol{C}$ is the graph convolution matrix (e.g. $\boldsymbol{C} = \boldsymbol{D}^{-1/2}\boldsymbol{A}\boldsymbol{D}^{-1/2}$ in Vanilla GCN (Kipf & Welling, 2016)) and $\Psi_l$ is the feature transformation matrix. For the node classification problem, a classifier (e.g., an MLP) is usually appended to the output of a $k$-layer GNN model; while for link prediction, a similarity function is applied to the final output to compute the similarity between two node embeddings. As shown above, as GNNs inherently evolve the whole graph structure for convolution, it is notoriously challenging for scaling it up. It is worth noting that evolving sufficient neighbors during training is crucial for GNNs. Many studies (Duan et al., 2022; Zou et al., 2019) have shown that full-batch training generally outperforms mini-batch for GNNs on multi graph benchmarks. In practice, the lower borderline of batch size for training GNNs is usually thousands. However, when applying it to LMs, it makes the GNN-LM end-to-end training intractable, as a text occupies far more GPU memories than an embedding.

**Text Embeddings and Language Models.** Transforming text in low-dimensional dense embeddings serves as the upstream of textual graph representation learning and has been widely explored in the literature. To generate sentence embeddings with LMs, two commonly-used methods are $(i)$ average pooling (Reimers & Gurevych, 2019) by taking the average of all word embeddings along with attention mask and $(ii)$ taking the embedding of the `[CLS]` token (Devlin et al., 2018). With

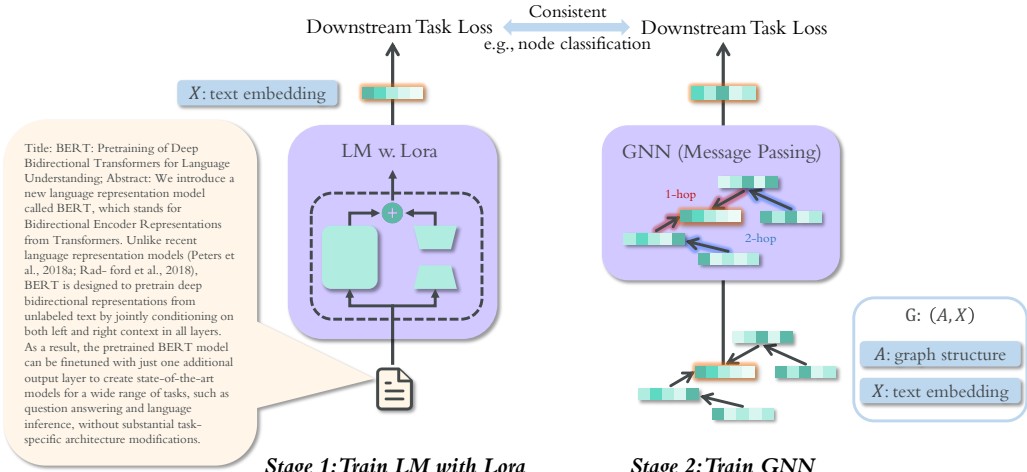

Figure 1: The overview of SimTeG. In *stage 1*, we train a LM with lora (Hu et al., 2022) and then generate the embeddings $X$ as the representation of text. In *stage 2*, we train a GNN on top of the embeddings $X$, along with the graph structure. The two stages are guided with consistent loss function, e.g., link prediction or node classification.

the development of pre-trained language models (Devlin et al., 2018; Liu et al., 2019), particular language models (Li et al., 2020; Reimers & Gurevych, 2019) for sentence embeddings have been proposed and shown promising results in various benchmarks (Muennighoff et al., 2022).

## 4   SIMTEG: METHODOLOGY

We propose an extremely simple two-stage training manner that decouples the training of $gnn(\cdot)$ and $lm(\cdot)$. We first finetune $lm$ on $\mathcal{T}$ with the downstream task loss:

$$Loss_{cls} = \mathcal{L}_\theta\big(\phi(lm(\mathcal{T})), \mathbf{Y}\big), \quad Loss_{link} = \mathcal{L}_\theta\big(\phi(lm(\mathcal{T}_{src}), lm(\mathcal{T}_{dst})), \mathbf{Y}\big), \tag{1}$$

where $\phi(\cdot)$ is the classifier (left for *node classification*) or similarity function (right for *link prediction*) and $\mathbf{Y}$ is the label. After finetuning, we generate node representations $X$ with the finetuned LM $\hat{lm}$. In practice, we follow Reimers & Gurevych (2019) to perform mean pooling over the output of the last layer of the LM and empirically find that such a strategy is more stable and converges faster than solely taking the <CLS> token embedding as representation (Zhao et al., 2022). In the second stage, we train $gnn$ on $(\mathbf{A}, \mathbf{X})$ with the same task. The corresponding loss is computed by replacing $lm(\mathcal{T})$ with $gnn(\mathbf{A}, \mathbf{X})$. The two stage is fully decoupled and one can take advantage of any existing GNN and LM models. We illustrate the two stages in Fig. 1 and the pseudo code is presented in Appendix A2.1.

**Regularization with PEFT.** When fully finetuning a LM, the inferred features are prone to overfit the training labels, which results in collapsed feature space and thus hindering the generalization in GNN training. Though PEFT was proposed to accelerate the finetuning process without loss of performance, in our two-stage finetuning stage, we empirically find PEFT (Hu et al., 2022; Houlsby et al., 2019; He et al., 2022) could alleviate the overfitting issue to a large extent and thus provide well-regularized node features. See Sec. 5.3 for empirical analysis. In this work, We take the popular PEFT method, lora (Hu et al., 2022), as the instantiation.

**Selection of LM.** As the revolution induced by LMs, a substantial number of valuable pre-trained LMs have been proposed. As mentioned before, when ablating graph structures of TG, the two fundamental tasks, *node classification* and *link prediction*, are simplified into two well-established NLP tasks, *text classification* and *text similarity (retrieval)*. Based on this motivation, we select LMs pretrained for information retrieval as the backbone of SimTeG. Concrete models are selected based on the benchmark MTEB[1] considering the model size and the performance on both retrieval and classification tasks. An ablation study regarding this motivation is presented in Sec. 5.4.

---

[1] https://huggingface.co/spaces/mteb/leaderboard

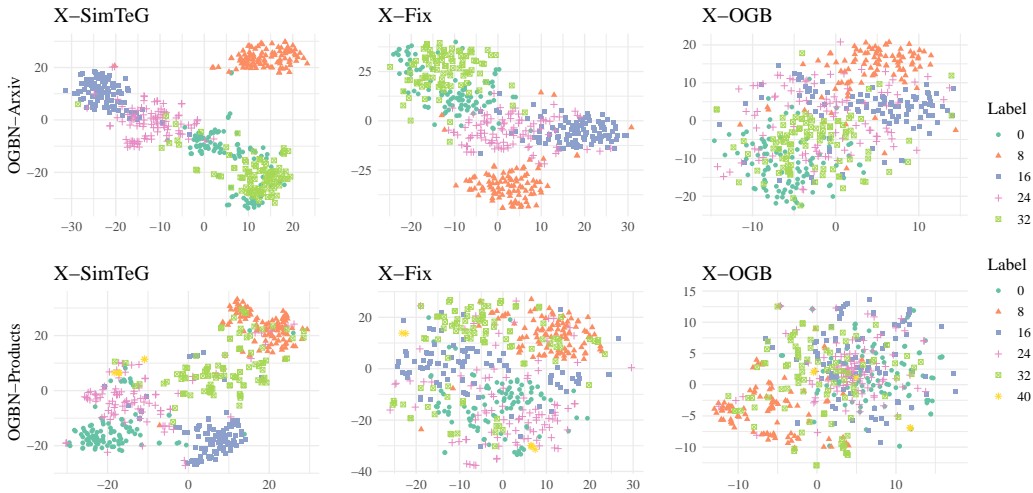

Figure 2: The two-dimensional feature space of $X$-SimTeG, $X$-Fix, and $X$-OGB for `OGBN-Arixv`, and `OGBN-Products`. $X$-SimTeG denotes the features generated by the finetuned LM. different values and shapes refer to different labels on the specific dataset. The feature values are computed by T-SNE. The LM backbone is e5-large (Wang et al., 2022).

**A Finetuned LM Provides A More Distinguishable Feature Space.** We plot the two-dimensional feature space computed by T-SNE (Van der Maaten & Hinton, 2008) of $X$-SimTeG, $X$-Fix (features generated by pretrained LM without finetuning), and $X$-OGB regarding labels on `OGBN-Arxiv` and `OGBN-Products` in Fig. 2. In detail, we randomly select 100 nodes each with various labels and use T-SNE to compute its two-dimensional features. As shown below, $X$-SimTeG has a significantly more distinguishable feature space as it captures more semantic information and is finetuned on the downstream dataset. Besides, we find that $X$-Fix is more distinguishable than $X$-OGB, which illustrates the inner semantic capture ability of LMs. Furthermore, in comparison with `OGBN-Arixv`, features in `OGBN-Products` is visually indifferentiable, indicating the weaker correlation between semantic information and task-specific labels. It accounts for the less improvement of SimTeG on `OGBN-Products` in Sec. 5.1.

## 5 EXPERIMENTS

In the experiments, we aim at answering three research questions as proposed in the introduction (Sec. 1). For a clear statement, we split and reformat them into the following research questions. **Q1:** How much could SimTeG generally improve the learning of GNNs on node classification and link prediction? **Q2:** Does X-SimTeG facilitate better convergence for GNNs? **Q3:** Is PEFT a necessity for LM finetuning stage? **Q4:** How sensitive is GNN training to the selection of LMs?

**Datasets.** Focusing on two fundamental tasks node classification and link prediction, we conduct experiments on three prestigious benchmarks: `OGBN-Arxiv` (Arxiv), `OGBN-Products` (Products), and `OGBL-Citation2` (Hu et al., 2020). The former two are for node classification while the latter one is for link prediction. For the former two, we follow the public split, and all text resources are provided by the officials. For the latter one, `OGBL-Citation2`, as no official text resources are provided, we take the intersection of it and another dataset `ogbn-papers100M` w.r.t. unified paper ids, which results in a subset of `OGBL-Citation2` with about 2.7M nodes. The public split is further updated according to this subset. In comparison, the original `OGBL-Citation2` has about 2.9M nodes, which is on par with the TG version, as the public valid and test split occupies solely 2% overall. As a result, we expect roughly consistent performance for methods on the TG version of `OGBL-Citation2` and the original one. We introduce the statistics of the three datasets in Table. A10 and the details in Appendix A2.2.

**Baselines.** We compare SimTeG with the official features $X$-OGB (Hu et al., 2020), which is the mean of word embeddings generated by skip-gram (Mikolov et al., 2013). In addition, for

node classification, we include another two SOTA methods: $X$-GIANT (Chien et al., 2021) and GLEM (Zhao et al., 2022). Particularly, $X$-* are methods are different at learning node embeddings and any GNN model could be applied in the downstream task for a fair comparison. To make things consistent, we denote our method as $X$-SimTeG without further specification.

**GNN Backbones.** Aiming at investigating the general improvement of SimTeG, for each dataset, we select two commonly-used baselines GraphSAGE and MLP besides one corresponding SOTA GNN models based on the official leaderboard[2]. For `OGBN-Arxiv`, we select RevGAT (Li et al., 2021); for `OGBN-Products`, we select SAGN+SCR (Sun et al., 2021; Zhang et al., 2021a); and for `ogbn-citation2`, we select SEAL (Zhang & Chen, 2018).

**LM Backbones.** For retrieval LM backbones, we select three popular LMs on MTEB (Muennighoff et al., 2022) leaderboard[3] w.r.t. model size and performance on classification and retrieval: *all-MiniLM-L6-v2* (Reimers & Gurevych, 2019), *all-roberta-large-v1* (Reimers & Gurevych, 2019), and *e5-large-v1* (Wang et al., 2022). We present the properties of the three LMs in Table. A12.

**Hyperparameter search.** We utilize *optuna* (Akiba et al., 2019) to perform hyperparameter search on all tasks. The search space for LMs and GNNs on all datasets is presented in Appendix A2.4.

### 5.1  **Q1**: HOW MUCH COULD SIMTEG *generally* IMPROVE THE LEARNING OF GNNS ON NODE CLASSIFICATION AND LINK PREDICTION?

In this section, we conduct experiments to show the superiority of SimTeG on improving the learning of GNNs on node classification and link prediction. The reported results are selected based on the validation dataset. We present the results based on *e5-large* backbone in Table. 1 and present the comprehensive results of node classification and link prediction with all the three selected backbones in Table A5 and Table A6. Specifically, in Table 1, we present two comparison metric $\Delta_{MLP}$ and $\Delta_{GNN}$ to describe the performance margin of (*SOTA GNN, MLP*) (*SOTA GNN, GraphSAGE*), respectively. The smaller the value is, even negative, the better the performance of simple models is. In addition, we ensemble the GNNs with multiple node embeddings generated by various LMs and text resources on `OGBN-Arxiv` and show the results in Table 2. We find several interesting observations as follows.

**Observation 1: SimTeG generally improves the performance of GNNs on node classification and link prediction by a large margin.** As shown in Table 1, SimTeG consistently outperforms the original features on all datasets and backbones. Besides, in comparison with $X$-GIANT, a LM pretraining method that utilizes the graph structures, SimTeG still achieves better performance on `OGBN-Arxiv` with all backbones and on `OGBN-Products` with GraphSAGE, which further indicates the importance of text attributes per se.

**Observation 2: ($X$-*SimTeG* + *GraphSAGE*) consistently outperforms ($X$-*OGB* + *SOTA GNN*) on all the three datasets.** This finding implies that the incorporation of advanced text features can bypass the necessity of complex GNNs, which is why we perceive our method to be *frustratingly* simple. Furthermore, when replacing GraphSAGE with the corresponding SOTA GNN in $X$-SimTeG, although the performance is improved moderately, this margin of improvement is notably smaller compared to the performance gap on $X$-OGB. Particularly, we show that the simple 2-layer GraphSAGE achieves comparable performance with the dataset-specific SOTA GNNs. Particularly, on `OGBN-Arxiv`, GraphSAGE achieves $76.84\%$, taking the ***4-th*** place in the corresponding leaderboard (by 2023-08-01). Besides, on `OGBL-Citation2`, GraphSAGE even outperforms the SOTA GNN method SEAL on Hits@3.

**Observation 3: With additional text attributes, SimTeG with Ensembling achieves *new SOTA performance* on `OGBN-Arxiv`.** We further demonstrate the effectiveness of SimTeG by ensembling the node embeddings generated by different LMs and texts. For text, we use both the original text provided by Hu et al. (2020) and the additional text attributes[4] provided by He et al. (2023), which is generated by ChatGPT. For LMs, we use both e5-large and all-roberta-large-v1. We train GraphSAGE or RevGAT on those node embeddings generated by various LMs and texts, and make the final predictions with weighted ensembling (taking the weighted average of all predictions). As

---

[2] https://ogb.stanford.edu/docs/leader_nodeprop

[3] https://huggingface.co/spaces/mteb/leaderboard

[4] It is worth noting that as GPT-4 used by He et al. (2023) does not release their training recipe, we do not know whether the arxiv papers are included during training, which may lead to a label leakage problem.

Table 1: The performance of SOTA GNN, GraphSAGE and MLP on `OGBN-Arxiv`, `OGBN-Products`, `OGBL-Citation2`, which are averaged over 10 runs (Please note the we solely train LM once to generate the node embeddings). The results of GLEM is from the orignal paper. We **bold** the best results w.r.t. the same GNN backbone and red color the smallest $\Delta_{MLP}$ and $\Delta_{GNN}$.

| Dataset | Metric | Method | SOTA GNN | A 2-layer Simple MLP / GNN | | | |
| --- | --- | --- | --- | --- | --- | --- | --- |
| | | | RevGAT | MLP | $\Delta_{MLP}$ | GraphSAGE | $\Delta_{GNN}$ |
| Arxiv | Acc. (%) | $X$-OGB | $74.01 \pm 0.29$ | $47.73 \pm 0.29$ | 25.24 | $71.80 \pm 0.20$ | 3.40 |
| | | $X$-GIANT | $75.93 \pm 0.22$ | $71.08 \pm 0.22$ | 4.85 | $73.70 \pm 0.09$ | 2.23 |
| | | GLEM | $76.97 \pm 0.19$ | - | - | $75.50 \pm 0.24$ | 1.47 |
| | | $X$-SimTeG | $\mathbf{77.04 \pm 0.13}$ | $\mathbf{74.06 \pm 0.13}$ | 2.98 | $\mathbf{76.84 \pm 0.34}$ | 0.20 |

| Dataset | Metric | Method | SOTA GNN | A 2-layer Simple MLP / GNN | | | |
| --- | --- | --- | --- | --- | --- | --- | --- |
| | | | SAGN+SCR | MLP | $\Delta_{MLP}$ | GraphSAGE | $\Delta_{GNN}$ |
| Products | Acc. (%) | $X$-OGB | $81.82 \pm 0.44$ | $50.86 \pm 0.26$ | 30.96 | $78.81 \pm 0.23$ | 3.01 |
| | | $X$-GIANT | $86.12 \pm 0.34$ | $\mathbf{77.58 \pm 0.24}$ | 8.54 | $82.84 \pm 0.29$ | 3.28 |
| | | GLEM | $\mathbf{87.36 \pm 0.07}$ | - | - | $83.16 \pm 0.19$ | 4.20 |
| | | $X$-SimTeG | $85.40 \pm 0.28$ | $76.73 \pm 0.44$ | 8.67 | $\mathbf{84.59 \pm 0.44}$ | 0.81 |

| Dataset | Metric | Method | SOTA GNN | A 2-layer Simple MLP / GNN | | | |
| --- | --- | --- | --- | --- | --- | --- | --- |
| | | | SEAL | MLP | $\Delta_{MLP}$ | GraphSAGE | $\Delta_{GNN}$ |
| Citation2 | MRR (%) | $X$-OGB | $86.14 \pm 0.40$ | $25.44 \pm 0.01$ | 60.70 | $77.31 \pm 0.90$ | 8.83 |
| | | $X$-SimTeG | $\mathbf{86.66 \pm 1.21}$ | $\mathbf{72.90 \pm 0.14}$ | 13.76 | $\mathbf{85.13 \pm 0.73}$ | 1.53 |
| | Hits@3 (%) | $X$-OGB | $90.92 \pm 0.32$ | $28.22 \pm 0.02$ | 62.70 | $85.56 \pm 0.69$ | 5.36 |
| | | $X$-SimTeG | $\mathbf{91.42 \pm 0.19}$ | $\mathbf{80.55 \pm 0.13}$ | 10.87 | $\mathbf{91.62 \pm 0.87}$ | -0.20 |

Table 2: The performance of GraphSAGE and RevGAT trained on `OGBN-Arxiv` with additional text attributes provided by He et al. (2023). LMs for ensembling are e5-large and all-roberta-large-v1. We select the top-3 SOTA methods from the leaderboard of `OGBN-Arxiv` (accessed on *2023-07-18*) for comparison and gray color our results (reported over 10 runs).

| Rank | Method | GNN Backbone | Valid Acc. (%) | Test Acc. (%) |
| --- | --- | --- | --- | --- |
| 1 | TAPE + SimTeG (ours) | RevGAT | $\mathbf{78.46 \pm 0.04}$ | $\mathbf{78.03 \pm 0.07}$ |
| 2 | TAPE (He et al., 2023) | RevGAT | $77.85 \pm 0.16$ | $77.50 \pm 0.12$ |
| 3 | TAPE + SimTeG (Ours) | GraphSAGE | $77.89 \pm 0.08$ | $77.48 \pm 0.11$ |
| 4 | GraDBERT (Mavromatis et al., 2023) | RevGAT | $77.57 \pm 0.09$ | $77.21 \pm 0.31$ |
| 5 | GLEM (Zhao et al., 2022) | RevGAT | $77.46 \pm 0.18$ | $76.94 \pm 0.25$ |

shown in Table 2, with RevGAT, we achieve new SOTA performance on `OGBN-Arxiv` with 78.03% test accuracy, more than 0.5 % higher than the previous SOTA performance (77.50%) achieved by He et al. (2023). It further validates the importance of text features and the effectiveness of SimTeG.

**Observation 4: Text attributes are unequally important for different datasets.** As shown in Table 1, we compute $\Delta_{MLP}$ which is the performance gap between MLP and SOTA GNNs. Empirically, this value indicates the importance of text attributes on the corresponding dataset, as MLP is solely trained on the texts (integrated with SOTA LMs) while SOTA GNN additionally takes advantage of graph structures. Therefore, approximately, the less $\Delta_{MLP}$ is, the more important text attributes are. As presented in Table 1, $\Delta_{MLP}$ on `OGBN-Arxiv` is solely 2.98, indicating the text attributes are more important, in comparison with the ones in `OGBN-Products` and `OGBL-Citation2`. This empirically indicates why the performance of SimTeG in `OGBN-Products` does not perform as well as the one in `OGBN-Arxiv`. We show a sample of text in `OGBN-Arxiv` and `OGBN-Products` respectively in Appendix A2.2. We find that the text in `OGBN-products` resembles more a bag of words, which account for the less improvement when using LM features.

## 5.2 Q2: DOES X-SIMTEG FACILITATE BETTER CONVERGENCE FOR GNNS?

Towards a comprehensive understanding of the effectiveness of SimTeG, we further investigate the convergence of GNNs with SimTeG. We compare the training convergence and the corresponding validation performance of GNNs trained on SimTeG, $X$-OGB, and $X$-FIX, where $X$-FIX denotes

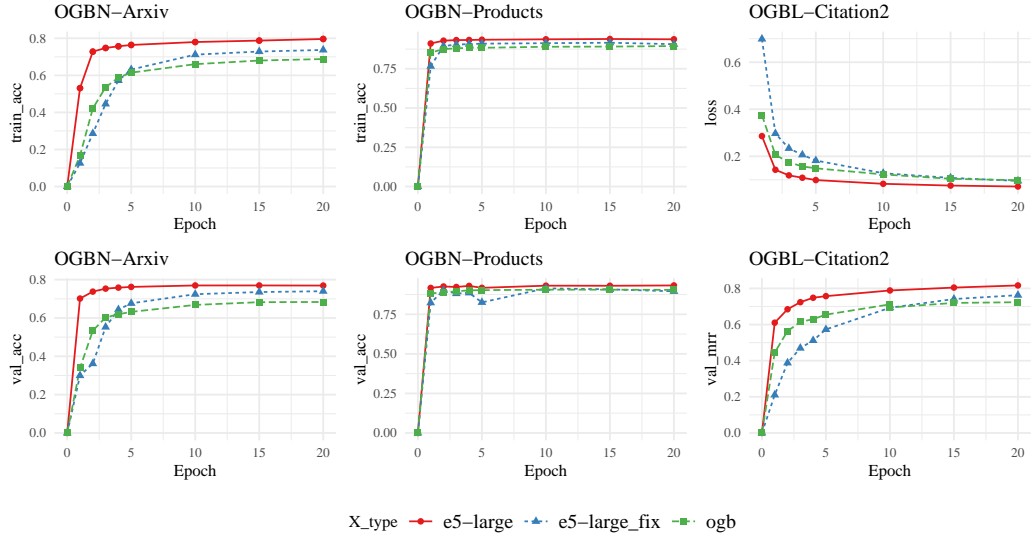

Figure 3: Training convergence and validation results of GNNs with $X$-SimTeG, $X$-OGB, and $X$-FIX. The LM backbone is e5-large. The learning rate and batch size are consistent.

Table 3: The training results of finetuning LM (*LM stage*) and further training GNN on top of derived features (*GNN stage*). We compare the results of PEFT (SimTeG) with full-finetuning ($X$-FULL). The LM backbone is e5-large and the GNN backbone is GraphSAGE. We **bold** the better results on each comparison. $\Delta_{overfit}$ computes (Train Acc. - Test Acc.) to measure the overfitting.

| datasets | Stage | X_type | Train Acc. | Valid Acc | Test Acc. | $\Delta_{overfit}$ |
|---|---|---|---|---|---|---|
| Arxiv | LM | $X$-FULL | 82.33 | 75.85 | **74.77** | 7.56 |
| | | $X$-SimTeG | 75.72 | 75.40 | 74.31 | 1.41 |
| | GNN | $X$-FULL | 84.39 | 76.73 | 75.28 | 9.11 |
| | | $X$-SimTeG | 79.37 | 77.47 | **76.85** | 2.52 |
| Products | LM | $X$-FULL | 95.46 | 91.70 | **78.70** | 16.76 |
| | | $X$-SimTeG | 89.45 | 88.85 | 77.81 | 11.64 |
| | GNN | $X$-FULL | 96.42 | 93.18 | 81.80 | 14.62 |
| | | $X$-SimTeG | 95.37 | 93.57 | **84.58** | 10.79 |

the node embeddings generated by the pretrained LMs without finetuning. The illustration is placed in Fig. 3. It is worth noting that we use the training accuracy on `OGBN-Arxiv` and `OGBN-Products` to denote their convergence since we utilize *label smoothing* during training which make the training loss not directly comparable on them. Based on Fig. 3, we have the following observation:

**Observation 5: SimTeG moderately speeds up and stabilizes the training of GNNs.** As shown in Fig. 3, GNNs with SimTeG generally converge faster than the ones with $X$-OGB and $X$-FIX. With SimTeG, GraphSAGE could converge within 2 epochs on `OGBN-Arxiv` and `OGBN-Products`. In contrast, training on the features directly generated by the pretrained LMs (i.e., $X$-FIX) converges much slower, even slower than one of $X$-OGB (possibly due to a larger hidden dimension). This further indicates the benefits of SimTeG.

### 5.3 **Q3**: IS PEFT A NECESSITY FOR LM FINETUNING STAGE?

In this ablation study, we analyze the effectiveness of PEFT for LM finetuning stage in SimTeG. Besides the accelerating finetuning, we also find notable contribution of PEFT to the effectiveness. We summarize the training, validation, and test accuracy of two stages: LM finetuning stage and GNN training stage. The results of node classification are presented in Table 3.

**Observation 6: PEFT could significantly alleviate the overfitting problem during finetuning LM and further facilitate the training of GNNs with regularized features.** As shown in Table 3, due to the excessively strong learning capacity of LMs, finetuning LMs on the downstream task causes a severe overfitting problem. Although full-finetuning outperforms PEFT in LM stage, training GNNs

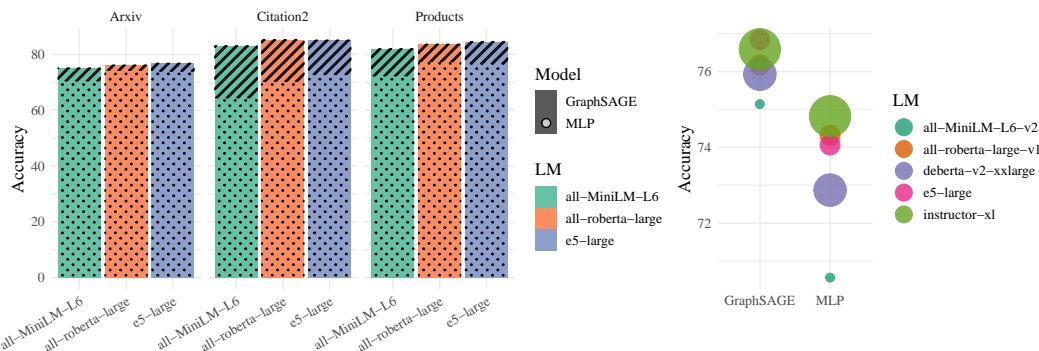

Figure 4: *(Left)*: The performance of GraphSAGE trained on SimTeG with different LM backbones on three datasets. *(Right)*: Comparative analysis of GNN's performance on `OGBN-Arxiv` using LMs of various sizes, indicated by the bubble size.

Table 4: The performance of Graph and MLP trained on SimTeG backed with all-roberta-large-v1 and roberta-large, which have the same model architecture. we **bold** the best results for each comparison in $X$-Fix and $X$-SimTeG. all results are reported based on 10 runs.

| datasets | Metric | X_type | $X$-Fix | | $X$-SimTeG | |
|---|---|---|---|---|---|---|
| | | LM Backbone | roberta-large | all-roberta-large-v1 | roberta-large | all-roberta-large-v1 |
| Arxiv | Acc. | MLP | 61.15 ± 0.83 | **72.58 ± 0.25** | 71.55 ± 0.24 | **74.32 ± 0.12** |
| | | GraphSAGE | 72.15 ± 0.59 | **75.51 ± 0.23** | 75.48 ± 0.16 | **76.18 ± 0.37** |
| Products | Acc. | MLP | 68.14 ± 0.23 | **70.10 ± 0.08** | 78.45 ± 0.14 | **77.48 ± 0.19** |
| | | GraphSAGE | 77.65 ± 0.34 | **82.38 ± 0.60** | 83.56 ± 0.21 | **83.68 ± 0.32** |
| Citation2 | MRR | MLP | 00.20 ± 0.01 | **70.12 ± 0.12** | 63.15 ± 0.20 | **72.90 ± 0.14** |
| | | GraphSAGE | 79.71 ± 0.27 | **83.20 ± 0.40** | 84.37 ± 0.34 | **85.13 ± 0.73** |

on the derived features gains notably less improvement. In contrast, PEFT could significantly mitigate the overfitting issue according to $\Delta_{overfit}$ in LM finetuning stage and assist the training of GNNs with regularized features to gain considerable improvement compared with full-finetuning.

### 5.4 Q4: HOW SENSITIVE IS GNN TRAINING TO THE SELECTION OF LMS?

In this experiment, we investigate the effects of the selection of LMs. In detail, we aim at answering: Is the training of GNN sensitive to the selection of LMs? If so, what is the underlying factors? We conduct experiments on multiple LM backbones and have the following observations.

**Observation 7: GNN's training is moderately sensitive to the selection of LMs.** We select three retrieval LMs based on their rank in MTEB leaderboard in terms of the classification and retrieval performance. Interestingly, based on the leaderboard, the performance ranking is *e5-large > all-roberta-large-v1 > all-MiniLM-L6-v2*, which is consistent with their overall performance in left subfigure Figure 4 and Table A9. Based on the right subfigure of Figure 4, we find that the performance of downstream GNN is not closely correlated with the LM size, but probably with ability to generate representative text embeddings. To further validate this, we perform an ablation study regarding the comparison between a pretrained LM and the same LM finetuned for retrieval tasks. The results are shown in Table 4. We observe that given the same architecture, the models specifically finetuned for retrieval tasks (*all-roberta-large-v1*) generally perform better on tasks of TG representation learning.

## 6 CONCLUSION

In this work, we propose a frustratingly simple approach SimTeG for TG representation learning. We show that with a parameter-efficiently finetuned LM on the same downstream task first, a simple two-layer GraphSAGE trained on the generated node embeddings can achieve on-par state-of-the-art (SOTA) performance on `OGBN-Arxiv` (77.48 %). It indicates that when incorporating advaced text features, one can bypass the necessity of using complex GNN architectures and the combination of *LM + Simple GNN* is capable of achieving satisfactory results on graph tasks including node classification and link prediction.

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

## A1 MORE EXPERIMENT RESULTS

### A1.1 COMPREHENSIVE RESULTS OF MAIN EXPERIMENTS

Table A5: Node Classification Accuracy of **X**-SimTeG on ogbn-arxiv (Arxiv) and ogbn-products (Products). All reported results are averaged over 10 runs in the format of mean $\pm$ std. We red color the best results and blue color the runner-ups with the same GNN backbone. $\uparrow$ (%) denotes the improvement of **X**-SimTeG over the original feature $X$-OGB. $\Delta_{MLP}$ and $\Delta_{GNN}$ denotes the extreme value difference among all methods (including MLP) and GNNs, respectively.

| Datasets | GNN | Acc. (%) | Baselines | | | **X**-SimTeG | | | | | |
|---|---|---|---|---|---|---|---|---|---|---|---|
| | | | **X**-OGB | **X**-GIANT | GLEM[a] | MiniLM-L6 | $\uparrow$ (%) | e5-large | $\uparrow$ (%) | roberta-large | $\uparrow$ (%) |
| | MLP | val | $49.14 \pm 0.27$ | $72.02 \pm 0.16$ | - | $71.59 \pm 0.07$ | 22.45 | $75.08 \pm 0.09$ | 26.66 | $74.80 \pm 0.07$ | 25.66 |
| | | test | $47.73 \pm 0.29$ | $71.08 \pm 0.22$ | - | $70.56 \pm 0.09$ | 22.83 | $74.06 \pm 0.13$ | 26.33 | $74.32 \pm 0.12$ | 26.59 |
| | GraphSAGE | val | $72.80 \pm 0.18$ | $74.58 \pm 0.20$ | $76.45 \pm 0.05$ | $75.92 \pm 0.17$ | 3.12 | $77.47 \pm 0.14$ | 4.67 | $76.86 \pm 0.13$ | 4.06 |
| | | test | $71.80 \pm 0.20$ | $73.70 \pm 0.09$ | $75.50 \pm 0.24$ | $75.14 \pm 0.30$ | 3.34 | $76.84 \pm 0.34$ | 5.04 | $76.18 \pm 0.37$ | 4.38 |
| Arxiv | GAMLP | val | $71.49 \pm 0.41$ | $76.36 \pm 0.09$ | $76.95 \pm 0.14$ | $76.75 \pm 0.11$ | 5.26 | $77.90 \pm 0.12$ | 6.41 | $77.57 \pm 0.15$ | 6.08 |
| | | test | $70.61 \pm 0.52$ | $75.26 \pm 0.15$ | $75.62 \pm 0.23$ | $75.46 \pm 0.17$ | 4.85 | $76.92 \pm 0.10$ | 6.31 | $76.72 \pm 0.19$ | 6.11 |
| | SAGN | val | $72.74 \pm 0.39$ | $75.76 \pm 0.21$ | - | $76.84 \pm 0.08$ | 4.10 | $78.03 \pm 0.05$ | 5.29 | $77.63 \pm 0.16$ | 4.89 |
| | | test | $71.76 \pm 0.41$ | $74.39 \pm 0.38$ | - | $75.50 \pm 0.23$ | 3.74 | $76.85 \pm 0.12$ | 5.09 | $76.59 \pm 0.17$ | 4.83 |
| | RevGAT | val | $75.10 \pm 0.15$ | $76.97 \pm 0.08$ | $77.49 \pm 0.17$ | $76.86 \pm 0.24$ | 1.76 | $77.68 \pm 0.07$ | 2.58 | $76.32 \pm 0.18$ | 1.22 |
| | | test | $74.01 \pm 0.29$ | $75.93 \pm 0.22$ | $76.97 \pm 0.19$ | $75.96 \pm 0.21$ | 1.95 | $77.04 \pm 0.13$ | 3.03 | $75.88 \pm 0.58$ | 1.87 |
| | $\Delta_{MLP}/\Delta_{GNN}$ | | 25.24 / 3.40 | 4.85 / 2.23 | - | 5.40 / 0.82 | - | 2.98 / 0.20 | - | 2.40 / 0.84 | - |
| | MLP | val | $63.44 \pm 0.30$ | $89.67 \pm 0.07$ | - | $86.82 \pm 0.02$ | 23.38 | $88.75 \pm 0.04$ | 25.31 | $90.01 \pm 0.03$ | 26.57 |
| | | test | $50.86 \pm 0.26$ | $77.58 \pm 0.24$ | - | $72.36 \pm 0.12$ | 21.50 | $76.73 \pm 0.44$ | 25.87 | $77.48 \pm 0.19$ | 26.62 |
| Products | GraphSAGE | val | $90.03 \pm 0.08$ | $93.49 \pm 0.09$ | $93.84 \pm 0.12$ | $93.49 \pm 0.08$ | 3.46 | $93.57 \pm 0.20$ | 3.54 | $93.34 \pm 0.09$ | 3.31 |
| | | test | $78.81 \pm 0.23$ | $82.84 \pm 0.29$ | $83.16 \pm 0.19$ | $82.04 \pm 0.57$ | 3.23 | $84.59 \pm 0.44$ | 5.78 | $83.68 \pm 0.32$ | 4.87 |
| | SAGN+SCR | val | $91.83 \pm 0.24$ | $94.04 \pm 0.12$ | $94.00 \pm 0.03$ | $92.89 \pm 0.07$ | 1.06 | $94.12 \pm 0.10$ | 2.29 | $94.13 \pm 0.12$ | 2.30 |
| | | test | $81.82 \pm 0.44$ | $86.12 \pm 0.34$ | $87.36 \pm 0.07$ | $82.43 \pm 0.40$ | 0.61 | $85.40 \pm 0.28$ | 3.58 | $85.23 \pm 0.32$ | 3.41 |
| | $\Delta_{MLP}/\Delta_{GNN}$ | | 30.96 / 3.01 | 8.54 / 3.28 | - | 10.07 / 0.39 | - | 8.67 / 0.81 | - | 7.75 / 1.55 | - |

[a] results are from the original papers.

Table A6: Link prediction results on *OGBL-Citation2-2.7M* (Citation2). All reported results are averaged over 10 runs. We red color the best results and blue color the runner-ups with the same GNN backbone. $\uparrow$ (%) denotes the improvement of **X**-SimTeG over the original feature $X$-OGB. $\Delta_{MLP}$ and $\Delta_{GNN}$ denotes the margin of (MLP, SEAL) and (GraphSAGE, SEAL), respectively. We use blue color denoting the negative values and red denoting positive. Specifically, in the context of $\Delta$, positives indicate MLP/GraphSAGE performs better than SEAL.

| Metrics | GNN | Split | Baselines | $X$-SimTeG | | | | | |
|---|---|---|---|---|---|---|---|---|---|
| | | | $X$-OGB | MiniLM-L6 | $\uparrow$ (%) | roberta-large | $\uparrow$ (%) | e5-large | $\uparrow$ (%) |
| | MLP | val | $25.37 \pm 0.09$ | $64.56 \pm 0.15$ | 39.19 | $70.20 \pm 0.19$ | 44.83 | $72.79 \pm 0.17$ | 47.42 |
| | | test | $25.44 \pm 0.01$ | $64.49 \pm 0.18$ | 39.05 | $70.32 \pm 0.22$ | 44.88 | $72.90 \pm 0.14$ | 47.46 |
| *MRR* | GraphSAGE | val | $77.40 \pm 0.88$ | $83.13 \pm 0.72$ | 5.73 | $85.27 \pm 0.78$ | 7.87 | $85.20 \pm 0.69$ | 7.80 |
| | | test | $77.31 \pm 0.90$ | $83.09 \pm 0.75$ | 5.78 | $85.29 \pm 0.70$ | 7.98 | $85.13 \pm 0.73$ | 7.82 |
| | SEAL | val | $87.21 \pm 0.03$ | $88.33 \pm 0.30$ | 1.12 | $88.29 \pm 0.45$ | 1.08 | $88.56 \pm 0.38$ | 1.35 |
| | | test | $86.14 \pm 0.40$ | $86.69 \pm 0.43$ | 0.55 | $87.02 \pm 0.46$ | 0.88 | $86.66 \pm 1.21$ | 0.52 |
| | $\Delta_{MLP}/\Delta_{GNN}$ | | -60.70 / -8.83 | -22.20 / -3.60 | - | -16.70 / -1.73 | - | -13.76 / -1.53 | - |
| | MLP | val | $15.04 \pm 0.09$ | $52.29 \pm 0.18$ | 37.25 | $59.46 \pm 0.19$ | 44.42 | $62.21 \pm 0.23$ | 47.17 |
| | | test | $15.11 \pm 0.06$ | $52.18 \pm 0.25$ | 37.07 | $59.66 \pm 0.26$ | 44.55 | $62.31 \pm 0.19$ | 47.20 |
| *Hits@1* | GraphSAGE | val | $67.28 \pm 1.20$ | $74.83 \pm 1.02$ | 7.55 | $77.98 \pm 1.20$ | 10.70 | $77.73 \pm 0.89$ | 10.45 |
| | | test | $67.09 \pm 1.25$ | $74.79 \pm 1.10$ | 7.70 | $77.99 \pm 0.89$ | 10.90 | $77.66 \pm 0.91$ | 10.57 |
| | SEAL | val | $82.76 \pm 0.14$ | $84.35 \pm 0.42$ | 1.59 | $84.25 \pm 0.79$ | 1.49 | $84.70 \pm 0.58$ | 1.94 |
| | | test | $81.74 \pm 0.46$ | $81.40 \pm 0.96$ | -0.34 | $82.34 \pm 0.79$ | 0.60 | $81.15 \pm 2.04$ | -0.59 |
| | $\Delta_{MLP}/\Delta_{GNN}$ | | -66.63 / -14.65 | -29.22 / -6.61 | - | -22.68 / -4.35 | - | -18.84 / -3.39 | - |
| | MLP | val | $28.06 \pm 0.10$ | $72.60 \pm 0.16$ | 44.54 | $77.56 \pm 0.23$ | 49.50 | $80.42 \pm 0.15$ | 52.36 |
| | | test | $28.22 \pm 0.02$ | $72.62 \pm 0.19$ | 44.40 | $77.66 \pm 0.24$ | 49.44 | $80.55 \pm 0.13$ | 52.33 |
| *Hits@3* | GraphSAGE | val | $85.54 \pm 0.69$ | $90.17 \pm 0.61$ | 4.63 | $91.55 \pm 0.98$ | 6.01 | $91.72 \pm 0.90$ | 6.18 |
| | | test | $85.56 \pm 0.69$ | $90.16 \pm 0.51$ | 4.60 | $91.57 \pm 1.10$ | 6.01 | $91.62 \pm 0.87$ | 6.06 |
| | SEAL | val | $91.36 \pm 0.44$ | $92.00 \pm 0.07$ | 0.64 | $92.15 \pm 0.19$ | 0.79 | $91.75 \pm 0.18$ | 0.39 |
| | | test | $90.92 \pm 0.32$ | $91.42 \pm 0.60$ | 0.50 | $91.52 \pm 0.56$ | 0.60 | $91.42 \pm 0.19$ | 0.50 |
| | $\Delta_{MLP}/\Delta_{GNN}$ | | -62.70 / -5.36 | -18.80 / -1.26 | - | -13.86 / 0.05 | - | -10.87 / 0.20 | - |
| | MLP | val | $46.73 \pm 0.14$ | $87.62 \pm 0.06$ | 40.89 | $89.80 \pm 0.20$ | 43.07 | $91.74 \pm 0.08$ | 45.01 |
| | | test | $46.59 \pm 0.11$ | $87.57 \pm 0.12$ | 40.98 | $89.66 \pm 0.14$ | 43.07 | $91.74 \pm 0.10$ | 45.15 |
| *Hits@10* | GraphSAGE | val | $94.29 \pm 0.19$ | $96.25 \pm 0.13$ | 1.96 | $96.61 \pm 0.12$ | 2.32 | $96.71 \pm 0.09$ | 2.42 |
| | | test | $94.37 \pm 0.17$ | $96.30 \pm 0.13$ | 1.93 | $96.64 \pm 0.12$ | 2.27 | $96.74 \pm 0.11$ | 2.37 |
| | SEAL | val | $94.59 \pm 0.14$ | $94.88 \pm 0.25$ | 0.29 | $95.08 \pm 0.12$ | 0.49 | $95.08 \pm 0.21$ | 0.49 |
| | | test | $93.90 \pm 0.49$ | $94.40 \pm 0.07$ | 0.50 | $93.95 \pm 0.37$ | 0.05 | $94.54 \pm 0.25$ | 0.64 |
| | $\Delta_{MLP}/\Delta_{GNN}$ | | -47.31 / -0.47 | -6.83 / 1.90 | - | -4.29 / 2.66 | - | -2.80 / 2.20 | - |

Table A7: The p values for two comparisons, SimTeG v.s. baseline (GIANT/OGB) and GraphSAGE v.s. SOTA GNN on three datasets. p value smaller than 0.05 means that SimTeG (or SOTA GNN) is significantly better than the baseline (GraphSAGE)

| Dataset | GNN | SimTeG | Baseline | P-Value | P <0.05 |
|---|---|---|---|---|---|
| OGBN-Arxiv | RevGAT | 77.04 | 75.93 | 7.77e-14 | True |
| | GraphSAGE | 76.84 | 73.70 | 4.11e-17 | True |
| OGBN-Products | SAGN+SLE | 85.40 | 86.12 | 4.15e-06 | True |
| | GraphSAGE | 84.59 | 82.84 | 9.01e-10 | True |
| OGBL-Citation2 | SEAL | 91.42 | 90.92 | 0.0023 | True |
| | GraphSAGE | 91.62 | 85.13 | 5.01e-12 | True |

| Dataset | Embeddings | GraphSAGE | SOTA GNN | P-Value | P <0.05 |
|---|---|---|---|---|---|
| OGBN-Arxiv | X-SimTeG | 76.84 | 77.04 | 0.0427 | True |
| | X-GIANT | 73.70 | 75.93 | 7.79e-22 | True |
| OGBN-Products | X-SimTeG | 84.59 | 85.40 | 8.74e-06 | True |
| | X-GIANT | 82.84 | 86.12 | 7.87e-17 | True |
| OGBL-Citation2 | X-SimTeG | 91.62 | 91.42 | 0.1380 | False |
| | X-OGB | 85.13 | 90.92 | 2.98e-15 | True |

Table A8: The results of more LM-involved methods. All results are averaged over 10 runs.

| Method | ogbn-arxiv | ogbn-products | ogbl-citation (MRR) |
|---|---|---|---|
| GraphFormer | $72.81 \pm 0.20$ | $74.72 \pm 0.16$ | $82.78 \pm 0.24$ |
| SimTeG + GraphSAGE | $76.84 \pm 0.34$ | $84.59 \pm 0.44$ | $85.13 \pm 0.73$ |
| SimTeG + SOTA GNN | $77.04 \pm 0.13$ | $85.40 \pm 0.28$ | $86.66 \pm 1.21$ |

### A1.2 P VALUE ANALYSIS

As shown in the upper sub-table in Table A7, all p values are lower than 0.05, indicating the significant improvement of SimTeG. It is worth noting that as the results of GLEM are reported by the original paper and we do not have the results for each individual experiment, we are not able to compute the corresponding p values. We do acknowledge that there is a subtle difference between SimTEG and GLEM and GLEM outperforms SimTEG on OGBN-Products. This phenomenon is discussed in our Observation 4 in Section 5.1 of the paper.

In addition, as shown in the bottom sub-table in Table A7, the p values of SimTeG are significantly smaller than the baseline embeddings. Specifically, the p values of SimTeG on `OGBN-Arxiv` and `OGBL-Citation2` are close or larger than 0.05. This further supports our key findings: in cooperation with advanced text embeddings, one can bypass the necessity of using complex GNN models.

### A1.3 COMPARISON WITH MORE LM-INVOLVED METHODS

The results of GraphFormer on OGBN-Arxiv and OGBN-Products are directly borrowed from the GLEM paper (Zhao et al., 2022). since the datasets and split are exactly the same. We run Graph-Former on ogbl-citation2 for 10 times and report the mean with std. For the hyperparameter setting, we use the default parameters, and the batch size is set to 100 to make it consistent with the reported results in GLEM. As shown in the table, SimTeG performs consistently better than GraphFormer. It is possibly because $(i)$ the GNN-nested architecture of GraphFormer solely allows 1-hop message passing, which limits the express ability of GNN models; $(ii)$ GraphFormer's implementation modifies the architecture code of BERT (Devlin et al., 2018) and cannot be easily extended to other SOTA embedding models nowadays.

Table A9: The performance of GraphSAGE and MLP trained on SimTeGwith different LM backbones. The former three LMs are finetuned with searched hyperparameters. For each row, we **bold** the best result and underline the runner-up. all results are reported based on 10 runs.

| datasets | Metric | LM backbone | all-MiniLM-L6-v2 | all-roberta-large-v1 | e5-large |
|----------|--------|-------------|------------------|----------------------|----------|
|          |        | #Params.    | 22M              | 355M                 | 335M     |
| Arxiv    | Acc.   | MLP         | $70.56 \pm 0.09$ | $74.32 \pm 0.12$     | $74.06 \pm 0.13$ |
|          |        | GraphSAGE   | $75.14 \pm 0.30$ | $\underline{76.18 \pm 0.37}$ | $\mathbf{76.84 \pm 0.34}$ |
| Products | Acc.   | MLP         | $72.36 \pm 0.12$ | $77.48 \pm 0.19$     | $76.73 \pm 0.44$ |
|          |        | GraphSAGE   | $82.04 \pm 0.57$ | $\underline{83.68 \pm 0.32}$ | $\mathbf{84.59 \pm 0.44}$ |
| Citation2 | MRR   | MLP         | $64.49 \pm 0.18$ | $70.32 \pm 0.22$     | $72.90 \pm 0.14$ |
|          |        | GraphSAGE   | $83.09 \pm 0.75$ | $\mathbf{85.29 \pm 0.70}$ | $\underline{85.13 \pm 0.73}$ |

## A1.4 MORE RESULTS OF ABLATION STUDIES

## A2 REPRODUCIBILITY STATEMENT

### A2.1 PSEUDO CODE OF SIMTEG

**Algorithm 1:** PyTorch-style code of SimTeG. Left: *node classification*; Right: *link prediction*.

```
# f_lm: language model wrapped with PEFT method, f_mlp: mlp model, f_gnn: gnn model
# inputs: A textual graph (adj_t (A), input_ids (T), att_mask (M)) and task-specific labels (Y)
# outputs: node representations

# Node Classification                     # Link Prediction
for T, M, Y in train_loader:              for (T_src, T_dst), (M_src, M_dst), Y in train_loader:
    X = f_lm(T, M)                            X_src, X_dst = f_lm((T_src, M_src), (T_dst, M_dst))
    logits = f_mlp(X)                         logits = f_mlp(X_src, X_dst)
    loss = CrossEntropyLoss(logits, Y)        loss = BCEWithLogitsLoss(logits, Y)
    loss.backward()                           loss.backward()
    lm_optimizer.step()                       lm_optimizer.step()

with torch.no_grad():                     with torch.no_grad():
    X = f_lm(T, M)                            X = lm(T, M)

f_mlp.reset_parameters()                  f_mlp.reset_parameters()
for A, X, Y in train_loader:              for A, (X_src, X_dst), Y in train_loader:
    X = f_gnn(A, X)                           X_src, X_dst = f_gnn(A, (X_src, X_dst))
    logits = f_mlp(X)                         logits = f_mlp(X_src, X_dst)
    loss = CrossEntropyLoss(logits, Y)        loss = BCEWithLogitsLoss(logits, Y)
    loss.backward()                           loss.backward()
    gnn_optimizer.step()                      gnn_optimizer.step()
```

### A2.2 DETAILS OF TG VERSION FOR THE THREE OGB DATASETS

In this section, we present the details of the TG version of `OGBN-Arxiv`, `OGBN-Products`, and `OGBL-Citation2`. The statistics of the three datasets are shown in Table A10 and the text resources are shown in Table A11.

Table A10: Statistics of `OGBN-Arxiv`, `OGBN-Products`, and `OGBL-Citation2-2.7M`

| Datasets | #Nodes | #Edges | Avg. Degree | #Task | Metric |
|----------|--------|--------|-------------|-------|--------|
| OGBN-Arxiv (Arxiv) | $169,343$ | $1,166,243$ | $13.7$ | node classification | Accuracy |
| OGBN-Products (Products) | $2,449,029$ | $61,859,140$ | $50.5$ | node classification | Accuracy |
| OGBL-Citation2-2.7M (Citation2) | $2,728,032$ | $27,731,705$ | $10.2$ | link prediction | MRR / Hits |

`OGBN-Arxiv`. `OGBN-Arxiv` is a directed academic graph, where node denotes papers and edge denotes directed citation. The task is to predict the category of each paper as listed in https://arxiv.org. For its TG version, we use the same split as Hu et al. (2020). The text for each node is its title and abstract. We concatenate them for each node with the format of "*title: {title}; abstract:*

*{abstract}*" as the corresponding node's text. For example, "title: multi view metric learning for multi view video summarization; abstract: Traditional methods on video summarization are designed to generate summaries for single-view video records; and thus they cannot fully exploit the redundancy in multi-view video records. In this paper, we present a multi-view metric learning framework for multi-view video summarization that combines the advantages of maximum margin clustering with the disagreement minimization criterion. ..."

`OGBN-Products`. `OGBN-Products` is a co-purchase graph, where node denotes a product on Amazon and an edge denotes the co-purchase relationship between two products. The task is to predict the category of each product (node classification). We follow the public split as Hu et al. (2020) and the text processing strategy of GLEM (Zhao et al., 2022). For each node, the corresponding text is its item description. For example, "My Fair Pastry (Good Eats Vol. 9)" "Disc 1: Flour Power (Scones; Shortcakes; Southern Biscuits; Salmon Turnovers; Fruit Tart; Funnel Cake; Sweet or Savory; Pte Choux) Disc 2: Super Sweets 4 (Banana Spitsville; Burned Peach Ice Cream; Chocolate Taffy; Acid Jellies; Peanut Brittle; Chocolate Fudge; Peanut Butter Fudge) ..."

`OGBL-Citation2-2.7M`. `OGBL-Citation2-2.7M` is a citation graph, where nodes denote papers and edges denote the citations. The task is to predict the missing citation among papers (link prediction). All papers are collected by the official from *Mircrosoft Academic Graph* whereas the text resources are not provided. Though MAG IDs for all papers are provided, we cannot find all corresponding text resources due to the close of MAG project [5]. Hence, we take an intersection of `OGBL-Citation2` and `OGBN-Papers100M` whose text resources are provided by the official, and build a subgraph, namely `OGBL-Citation2-2.7M`. It contains 93% nodes of `OGBL-Citation2` and offers a roughly on-par performance for baselines.

Table A11: The URLs of text resources for ogbn-arxiv, ogbn-products, and OGBL-Citation2.

| Dataset | Text Resource URL |
|---|---|
| OGBN-Arxiv | https://snap.stanford.edu/ogb/data/misc/ogbn_arxiv/titleabs.tsv.gz |
| OGBN-Products | https://drive.google.com/u/0/uc?id=1gsabsx8KR2N9jJz16jTcAOQASXsNuKnN&export=download |
| OGBL-Citation2-2.7M | https://drive.google.com/u/0/uc?id=19_hkbBUDFZTvQrMOoMbftuXhgz5LbIZY&export=download |

### A2.3 PROPERTIES OF LANGUAGE MODELS

Table A12: Properties of the selected LM backbones. Repository are hosted by huggingface.

| LM | #Params. | #Layers | #Hidden Dim. | Repository |
|---|---|---|---|---|
| all-MiniLM-L6-v2 | 23M | 6 | 384 | sentence-transformer/all-MiniLM-L6-v2 |
| all-roberta-large-v1 | 355M | 24 | 1024 | sentence-transformer/all-roberta-large-v1 |
| e5-large | 335M | 24 | 1024 | intfloat/e5-large |

### A2.4 HYPERPARAMETER SEARCH SPACE

For language models, we design the hyperparameter (HP) search space as in Table A13. Please note that for link prediction, the label smoothing factor is omitted. For HP searching, we utilize optuna (Akiba et al., 2019) to search the best HPs for each dataset and each model. For LMs, we take 10 trials. For GNNs, we take 20 trials. The final HP setting for LMs and GNNs are placed as shell scripts in our repository.

Table A13: The search space of LMs and GNNs.

| | LM | | | GNN | |
|---|---|---|---|---|---|
| hyperparameter | search space | type | hyperparameter | search space | type |
| learning rate | [1e-6, 1e-4] | continual | learning rate | [1e-4, 1e-2] | continual |
| weight decay | [1e-7, 1e-4] | continual | weight decay | [1e-7, 1e-4] | continual |
| label smoothing | [0.1, 0.7] | continual | label smoothing | [0.1, 0.7] | continual |
| header dropout | [0.1, 0.8] | continual | dropout | [0.1, 0.8] | continual |
| lora r | [1, 2, 4, 8] | descrete | num of layers | [2, 3, 4, 6, 8] | descrete |
| lora alpha | [4, 8, 16, 32] | descrete | | | |
| lora dropout | [0.1, 0.8] | continual | | | |

---

[5] https://www.microsoft.com/en-us/research/project/microsoft-academic-graph/

