# OpenReview forum: "SimTeG: A Frustratingly Simple Approach Improves Textual Graph Learning"
_ICLR.cc/2024/Conference — Submitted to ICLR 2024_

### Official Review · Reviewer_PJGj · 2023-10-23

**Soundness:** 3 good
**Presentation:** 3 good
**Contribution:** 2 fair
**Rating:** 5
**Confidence:** 5

**Summary:**

In this paper, the authors propose a very simple framework for learning on textual graphs. They conduct a two-step framework: 1) Finetune a language model on downstream tasks and obtain node representations; 2) Train a graph neural network model with the features from step 1 as node features. The authors then conduct experiments on three network datasets and perform model studies.

**Strengths:**

1. The paper is very clearly written and easy to follow.
2. The proposed framework is simple and useful.

**Weaknesses:**

1. Lack of comparison with existing works: GraphFormers [1], Patton [2]. There is another line of work [1,2] that tries to use only a language model to capture both semantic information and structure information in a textual graph. It would be more comprehensive to see how the performance comparison is between SimTeG and those methods.
2. Excitement of the findings and studies. I appreciate the authors’ detailed study of the two-stage pipeline. However, the finding is quite straightforward and not exciting enough to me. It is intuitive that the initial node feature vectors are very important and if a language model is trained on the downstream task first to generate the node feature vectors for the GNN methods, it will contribute to a very good performance.
3. Technical novelty. Correct me if I’m wrong, but this method can be seen as a single step for GLEM. The original GLEM involves iterative training of LM and GNN, while SimTeg contains only one round (LM training then GNN training). The performance comparison with GLEM is also very marginal regarding SOTA GNN.

[1] GraphFormers: GNN-nested Transformers for Representation Learning on Textual Graph. NeurIPs 2021.
[2] Patton: Language Model Pretraining on Text-Rich Networks. ACL 2023.

**Questions:**

Questions:
1. What is the performance comparison between SimTeG, GraphFormers [1], and Patton [2]?
2. See the second and third comments in the “Weakness” section.


Minor suggestions:
1. In Figure 2, please clarify which is referring to Arxiv and which is for products. Is X-SimTeG the embedding generated by LM (first stage) or GNN (second stage)?
2. Page 5, typo “how sensitive is GNN training sensitive to the selection of LMs?”

---

> ### Author Response · Authors · 2023-11-18
>
> Many thanks for the valuable comments and suggestions. We have accordingly updated our manuscript and provided responses as follows. All grammar errors and typos are corrected in the updated manuscripts.
>
> ## Comment #1: What is the performance comparison between SimTeG, GraphFormer and Patton?
>
> Thanks for the question. GraphFormer and Patton are valuable and related works. We have also included them in our related works. We report the results of GraphFormer on our three datasets as follows. For Patton, their implementation only supports their customized datasets and is not trivial to be adopted in our benchmarks. However, since patton and GraphFormer follow similar GNN-nested transformer architecture, it would be valuable to compare the performance of GraphFormer first.
>
> As shown in the following table, SimTeG performs consistently better than GraphFormer. It is possibly because 1) the GNN-nested architecture of GraphFormer solely allows 1-hop message passing, which limits the express ability of GNN models; 2) GraphFormer’s implementation modifies the architecture code of Bert and cannot be easily extended to other SOTA embedding models nowadays.
>
> We have added the GraphFormer results to Table A8 in the paper.
>
> | method | ogbn-arxiv | ogbn-products | ogbl-citation (MRR) |
> |---|---|---|---|
> |GraphFormer | 72.81 ± 0.20 | 74.72 ± 0.16 | 82.78 ± 0.24 |
> | SimTeG + GraphSAGE | 76.84 ± 0.34 | 84.59 ± 0.44 | 85.13 ± 0.73 |
> | SimTeG + SOTA GNN | 77.04 ± 0.13 | 85.40 ± 0.28 | 86.66 ± 1.21 |
>
> ## Comment #2: The finding is quite straightforward and not exciting enough to me. It is intuitive that the initial node features are important and if a language model is trained on the downstream task first to generate the node features, it will contribute to a very good performance.
>
> We provide a general response section above to clarify our motivation and contributions. We sincerely appreciate the reviewer checking it. Besides, we also moderately adjust our introduction section to make our motivation and argument clearer.
>
> We partially agree that “if a language model is trained on the downstream task first to generate the node features, it will contribute to a very good performance.” However several crucial questions remain open: (1) what is the extent of “very good”, especially with a simple GNN architecture? (2) Are advanced text embeddings good alternatives to complicated GNN architecture?
>
> We kindly disagree that “the findings are quite straightforward and not exciting enough”. We have summarized our contributions in the general response and hope it would address the concerns of the reviewer. Basically, the central argument is that advanced text features are good alternatives for complex GNNs. Validating this argument is beneficial for the application of GNNs in industry because industry applications of GNN architectures are limited by computational efficiency. To date, a notable exception is Pinsage [1], a GraphSAGE-based recommender system for Pinterest. If incorporating advanced text features could bypass the necessity of using complex GNN models, it would significantly boost the application of GNNs in industry.
>
> ## Comment #3: The technical novelty is not enough. This method can be seen as a single step for GLEM. The performance comparison with GLEM is also very marginal regarding SOTA GNN.
>
> As the statement of the general response and our abstract, we acknowledge that the proposed method is not technically novel, and this two-stage training has been explicitly or implicitly applied in various applications, e.g., GLEM [2]. Besides, related works [3][4] also utilize a similar mechanism. However, we state two key differences that distinguish our work.
>
> First, the motivation of our work is not to propose a novel framework that could generally outperform the other methods. Instead, we would like to utilize this simple two-stage training to demonstrate our key argument: *by incorporating advanced text features, one can bypass the necessity of using complex features.* This is mainly demonstrated by the performance improvement of a simple two-layer graphsage.
>
> Besides the key findings and motivation, other *unique empirical findings* in our work (e.g. The impact of PEFT in Section 5.3 and The impact of LM variants in Section 5.4) could also be generalized to any similar frameworks.
>
> [1] Ying, Rex, et al. "Graph convolutional neural networks for web-scale recommender systems." Proceedings of the 24th ACM SIGKDD international conference on knowledge discovery & data mining. 2018.
> [2] Zhao, Jianan, et al. "Learning on large-scale text-attributed graphs via variational inference." arXiv preprint arXiv:2210.14709 (2022).
> [3] He, Xiaoxin, et al. "Explanations as Features: LLM-Based Features for Text-Attributed Graphs." arXiv preprint arXiv:2305.19523 (2023).
> [4] Chen, Zhikai, et al. "Exploring the potential of large language models (llms) in learning on graphs." arXiv preprint arXiv:2307.03393 (2023).

---

> > ### Comment · Reviewer_PJGj · 2023-11-22
> >
> > Thank you so much for the detailed reply!
> >
> > 1. Baselines: For GraphFormers, a more fair comparison is using your SimTeG language model encoder (obtained after the first stage) to initialize the language model parameters in GraphFormers and conduct the second stage training. Otherwise, GraphFormers is using a general PLM and SimTeG is using a PLM adapted to your target corpus, which will result in an unfair comparison. On the other hand, if we treat the first step of SimTeG as pretraining the PLM, it is quite necessary to compare it with Patton which shares a similar philosophy. "their implementation only supports their customized datasets" is not a convincing reason, since you can always process your data to be a "customized datasets".
> >
> > 2. Novelty: I'm still not convinced by the novelty of this work. In text-attributed graph scenarios, node classification and link prediction just correspond to text classification and text matching. It is already demonstrated in the NLP domain that a more advanced text encoder can contribute to better performance in those tasks (in your case, your fine-tuned text encoder > GIANT fine-tune text encoder > bag-of-words). Although one can argue that in graph scenarios the graph structure information is important, if you improve the text representations, you improve the feature for GNN and everything will improve. Even related to the argument by the author "by incorporating advanced text features, one can bypass the necessity of using complex features", there are existing works having a similar exploration [1].
> >
> > [1] Vassilis et al. Efficient and effective training of language and graph neural network models. AAAI 2023.

---

### Official Review · Reviewer_wDRw · 2023-10-31

**Soundness:** 2 fair
**Presentation:** 3 good
**Contribution:** 2 fair
**Rating:** 3
**Confidence:** 4

**Summary:**

This paper studied a problem with textual graph learning by using the power of language models (LMs). The authors state that previous works focus on designing complex tasks or model structures for LMs on graph domains. However, the authors argue that it’s not necessary for such complexity, so they propose a simple and efficient method (SimTeG) for textual graph learning with LMs. Their proposed SimTeG improves the performance of GNNs on large-scale graph datasets for both node classification and link prediction tasks.

**Strengths:**

1. This paper studied an interesting problem about improving textual graph learning with language models (LMs). The author provides a thorough literature review in this domain.

2. Compared with previous methods for designing novel architecture or complex tasks, this paper proposes a simple and effective, where the authors perform Parameter-Efficient Fine-Tuning (PEFT) on a language model. Then, they utilize this fine-tuned language model to generate node representations from the text by omitting the top layer.

3. In this paper, the authors conduct extensive experiments on popular, large-scale graph datasets to evaluate both node classification and link prediction tasks. Their findings indicate that proficient language modeling can significantly enhance the performance of Graph Neural Network (GNN) models. Moreover, their straightforward approach demonstrated remarkable effectiveness in boosting performance.

**Weaknesses:**

1. The technical contribution of the paper appears to be limited, especially when considering the work of [1]. The core idea closely mirrors that of [1], which also leverages embeddings learned from a language model to enhance the learning of textual graph data via a variational expectation-maximization joint-training framework. The distinguishing factor in the proposed method is its two-step approach. However, I struggle to identify substantial contributions that differentiate it from [1].

2. The authors argue that prior methods have crafted intricate tasks and structures to bolster the performance of textual graph learning with LMs. However, existing methods like [1,2,3,4,5] are conceptually simple and their frameworks are straightforward. Moreover, their training processes do not necessitate significant modifications to the prevalent model architectures.

3. The paper's motivation is somewhat ambiguous. The authors predominantly focus on basic tasks, such as node classification and link prediction. Given that a rudimentary Graph Neural Network (GNN) can already yield satisfactory results for these tasks, the rationale for introducing a language model, which may be slower in inference and parameter-inefficient, is unclear. It might be more productive for the authors to highlight aspects like reduced inference time on test graph data or a more streamlined parameter set.

4. While the authors have undertaken link prediction experiments, there is a noticeable absence of comparisons with some of the state-of-the-art (SOTA) methods that incorporate LMs. It would be beneficial for them to showcase, for example, the performance of GLEM or other notable methods on the link prediction task. Such a comparison could further attest to the efficacy of their proposed model.

5. The review suggests compare the proposed method with GNNs which utilize bag-of-words feature not just the word-embedding feature in Ogbn-Arxiv.

References:

[1] Learning on Large-scale Text-attributed Graphs via Variational Inference. ICLR 2023

[2] Explanations as Features: LLM-based Features for Text-Attributed Graphs. Arxiv

[3] Node Feature Extraction by Self-supervised Multi-scale Neighborhood Prediction. ICLR 2022

[4] Exploring the Potential of Large Language Models (LLMs) in Learning on Graphs. Arxiv

[5] Natural Language is All a Graph Needs. Arxiv.

**Questions:**

1. On Ogbn-arxiv, SimTeG outperforms GLEM. From my understanding, GLEM can adaptively optimize the input embedding for GNNs, which will show better performance compared with SimTeG. Can the authors provide more discussions about this?
2. Could the authors provide further insights into the specific LM variants that can significantly enhance GNN models? For instance, it would be valuable to understand whether larger LM parameters or other factors play a substantial role in this improvement.

---

> ### Author Response · Authors · 2023-11-18
> **[1/2] Response**
>
> We are thankful for the reviewer’s constructive comments and valuable questions. We respond as follows and hope this could address the concerns. In addition to what follows, we also provide a general response above that may address the common concerns about our motivation and technical contributions to this work.
>
> ## Comment #1: The technical contribution of the paper appears to be limited, especially when considering GLEM.
>
> Thanks for the question. As mentioned in the general response and our abstract, the primary objective of this paper is not proposing a brand new and novel technique to achieve SOTA results on the leaderboard. Instead, we would like to use the proposed simple framework to validate our key argument: incorporating advanced text features could bypass the necessity of using SOTA GNNs. We refer the reviewer to look through our general responses above for a detailed introduction of our motivation and contributions.
>
> ## Comment #2: The authors argue that prior methods have crafted intricate tasks and structures to bolster the performance of textual graph learning with LMs. However, existing methods like [1,2,3,4,5] are conceptually simple and their frameworks are straightforward. Moreover, their training processes do not necessitate significant modifications to the prevalent model architectures.
>
> Thanks for the comments. In comparison with SimTeG, prior methods [1,2] and concurrent works [3,4,5] are relatively intricated in terms of their designed tasks or frameworks. For examples, GIANT [1] designs a graph-structure-related self-supervised task to generate text embeddings; GLEM [2] utilizes a joint training framework for GNNs and LMs to iteratively generate representative text embeddings; while the concurrent works [3,4,5] try to utilize LLMs to help generate more powerful text embeddings. Though showing promising results, the introduced tasks or techniques prevent us from directly validating our hypothesis from these works – whether incorporating advanced text features could bypass the necessity of using SOTA GNNs. As a result, we propose a simple framework to assess this argument.
>
> ## Comment #3: The paper’s motivation is somewhat ambiguous. Given that a rudimentary Graph Neural Network (GNN) can already yield satisfactory results for these tasks, the rationale for introducing a language model, which may be slower in inference and parameter-inefficient, is unclear.
>
> We kindly disagree that  “rudimentary GNN can already yield satisfactory for these tasks” – as shown in Table 1, rudimentary GNNs do not work well on text features generated by traditional NLP techniques such as Skip-gram, which is the official feature of OGB. These results motivate us to explore the effect of advanced text features from language models.
>
> In addition, as shown in different benchmarks, the SOTA GNN models are specialists for certain cases. For example, RevGAT is SOTA on OGBN-Arxiv; while SAGN + SLE dominates OGBN-Products; and SEAL-related works are the mainstream for some link prediction benchmarks. However, our results in Tables 1 and 2 show that with advanced text features, a two-layer GraphSAGE could generally perform well on various datasets and two different tasks. This may further ease the application of GNN models on various domains and tasks. We had a discussion about this on Observation 2 in Section 5.1 of the paper.
>
> ## Comment #4: While the authors have undertaken link prediction experiments, there is a noticeable absence of comparisons with some of the state-of-the-art (SOTA) methods that incorporate LMs.
>
> Thanks for the comments. We did not include the link prediction results of some SOTA methods as few previous works test their methods on link prediction tasks, and their provided code only implemented node classification and is non-trivial to be extended to link prediction tasks. To mitigate the reviewer’s concerns, we tried several related works and included the results of graphformer on our link prediction datasets. We provide it as follows and have added this baseline to Table A8 of the paper.
>
> | method | ogbn-arxiv | ogbn-products | ogbl-citation (MRR) |
> |---|---|---|---|
> |GraphFormer | 72.81 ± 0.20 | 74.72 ± 0.16 | 82.78 ± 0.24 |
> | SimTeG + GraphSAGE | 76.84 ± 0.34 | 84.59 ± 0.44 | 85.13 ± 0.73 |
> | SimTeG + SOTA GNN | 77.04 ± 0.13 | 85.40 ± 0.28 | 86.66 ± 1.21 |
>
> Following the general response, we note that competing for  SOTA results is not the focus of our paper. Instead, we aim to compare the performance of SOTA GNN and GraphSAGE with and without SimTeG, which could validate the central argument of our paper – incorporating advanced text features could bypass the necessity of using complex GNN architectures.

---

> ### Author Response · Authors · 2023-11-18
> **[2/2] Response**
>
> ## Comment #5: It is suggested to compare the proposed method with GNNs which utilize the bag-of-words feature.
>
> Thanks for the suggestions. As for our used datasets, the OGB does not provide the bag-of-words features, and we did not find GNN models and papers that utilize the bag-of-words feature on OGB datasets. Could you kindly provide more information or reference on this?
>
> Intuitively, we generally believe that the dense word embedding methods should work better than bag-of-words features because the former is recognized to be more representative on summarizing sentences and paragraphs.
>
> ## Comment #6: On OGBN-Arixv, SimTeG outperforms GLEM. GLEM can adaptively optimize the input embedding for GNNs, which will show better performance compared with SimTeG. Can the authors provide more discussions about this?
>
> Thanks for the question. SimTeG is similar to the first stage of GLEM, except for two key differences.
>
> SimTeG utilizes LoRA during the LM finetuning stage to alleviate the overfitting problem to generate more representative features.
> Second, SimTeG does not utilize pseudo labels during the GNN training stage. The pseudo labeling is crucial for GLEM and helps it iteratively optimize the embeddings. However, pseudo labeling does not help much on OGBN-Arxiv. As shown in the Figure 2 of GLEM paper, GNN already achieves the best performance on the first stage on OGBN-Arxiv dataset because of the ‘good prediction’ of LM. Instead, GLEM turns out to be more effective on more noisy datasets (e.g., OGBN-Products), where SimTeG fails to achieve on-par performance. We provided a related discussion in Observation 4 of Section 5.1 in the paper.
>
> As stated in the previous response, we believe that our empirical findings like utilizing LoRA in Section 5.3, and the study about impact of LM variants in Section 5.4 would also be beneficial to improve the performance of methods like GLEM.
>
>
> ## Comment #7: Could the authors provide further insights into the specific LM variants that can significantly enhance GNN models? For instance, it would be valuable to understand whether larger LM parameters or other factors play a substantial role in this improvement.
>
> Thanks for the question. We have discussed the influence of LM variants in section 5.4 and Table A7 of the original submission. We further elaborate on it and add new empirical results to make the discussion more complete. Basically, we summarize our findings below:
> the number of LM parameters is not substantial for the improvement. Including the results shown in our paper, we further test 2 more LMs that have different numbers of parameters, instructor-xl (1.5B) and deberta-v2-xxlarge (2.8B). The results on OGBN-Arxiv and statistics are shown in the following table and included in Figure 4 (right subfigure) of our updated manuscript. When we try to increase the number of parameters, the performance does not improve further. This is possibly because even though models with larger parameters could have better fine tuning node classification performance, the resulted text embedding is not correspondingly improved. Thus, we conjecture that the key factor for improving the performance of GNN is the quality of text embeddings, which, as shown as MTEB (https://huggingface.co/spaces/mteb/leaderboard), is not positively related to the number of parameters.
>
> | Metric | Model | all-MiniLM-L6-v2 | all-roberta-large-v1 | e5-large | deberta-v2-xxlarge | instructor-xl |
> |--------|----------|----------|-----------|--------|-------|---------|
> | #Params. |    | 22M       | 355M     | 335M      | 1.5B | 2.8B |
>  | Acc.   | MLP | 70.56 ± 0.09 | 74.32 ± 0.12 | 74.06 ± 0.13 |  72.87 ± 0.09   | 74.82 ± 0.08  |
> |          | GraphSAGE | 75.14 ± 0.30 | 76.18 ± 0.37 | 76.84 ± 0.34 |  75.93 ± 0.29  | 76.59 ± 0.36 |
>
> We performed an ablation study regarding the quality of text embeddings on Table A7 in our original paper, where we systematically compared two models with the exactly the same architecture, roberta-large and all-roberta-large-v1. The second one is the finetuned version of the first one using contrastive learning. The results show that for both LM finetuning stage and GNN training stage, all-roberta-large-v1 consistently outperforms roberta-large, indicating that language models specifically finetuned for text embedding tasks are more suitable for our cases.
>
>
> [1] Learning on Large-scale Text-attributed Graphs via Variational Inference. ICLR 2023
> [2] Node Feature Extraction by Self-supervised Multi-scale Neighborhood Prediction. ICLR 2022
> [3] Explanations as Features: LLM-based Features for Text-Attributed Graphs. Arxiv
> [4] Exploring the Potential of Large Language Models (LLMs) in Learning on Graphs. Arxiv
> [5] Natural Language is All a Graph Needs. Arxiv.

---

### Official Review · Reviewer_cWJD · 2023-11-04

**Soundness:** 3 good
**Presentation:** 3 good
**Contribution:** 3 good
**Rating:** 6
**Confidence:** 4

**Summary:**

This paper introduces SimTeG, a simple yet effective method for graph learning on textual graphs (where nodes have text attributes). The key idea of SimTeG is to fine-tune a pre-trained language model (PLM) for downstream tasks (e.g., node classification) and then take the PLM output representations as the input features to GNNs for the same tasks. Experimental results show that SimTeG significantly improves GNNs' performance on various graph benchmarks, where the authors examine various choices of GNN backbones and PLM backbones. Through extensive studies, the authors also obtain some meaningful observations, such as that PEFT is necessary when fine-tuning PLMs, that can guide future research in this direction.

**Strengths:**

+ Exploring the impact of PLMs on GNN learning and the importance of text attributes in various graph tasks is a meaningful task and has great potential given the recent breakthrough in large language models.

+ The proposed idea (i.e., training PLMs with LoRA + training GNNs) is simple but intuitive and well-motivated, which should be appreciated.

+ Experiments are quite comprehensive. Datasets from different domains (i.e., academic and e-commerce) are considered. Various GNN backbones and PLM backbones are examined, showing the generalizability of the proposed method.

+ Extensive analyses are conducted to obtain meaningful insights, such as the necessity of LoRA and the unequal importance of text attributes on different datasets.

**Weaknesses:**

- Statistical significance tests are missing. It is unclear whether the gaps between SimTeG and the baselines are statistically significant or not. In fact, some gaps in Tables 1-3 are subtle and unlikely significant given the reported standard deviation.

- An important baseline, GraphFormers [1], is not compared.

- Only LoRA is examined in the proposed method as the PEFT strategy. It is unclear whether other strategies, such as Prefix-Tuning and Adapter, can also help tackle the overfitting problem. If so, the observed necessity of PEFT would be strengthened.

[1] Graphformers: Gnn-nested transformers for representation learning on textual graph. NeurIPS'21.

**Questions:**

- Could you conduct statistical significance tests to compare SimTeG with the baselines and report the p-values?

- Could you report the performance of GraphFormers?

- Could you explore other PEFT strategies to check their effect on the overfitting problem?

---

> ### Author Response · Authors · 2023-11-18
>
> We are thankful for the reviewer’s constructive comments. We address the concerns both in the revised manuscript and in what follows.
>
> ## Comment #1: Statistical significance tests are missing. In fact, some gaps in Tables 1-3 are subtle and unlikely significant given the reported standard deviation.
>
> Thanks for the suggestions. We compute the p values for two comparisons, SimTeG v.s. baseline (GIANT/OGB) and GraphSAGE v.s. SOTA GNN on three datasets. As shown in the following second table where we compare between GraphSAGE and SOTA GNN, the p values of SimTeG are significantly smaller than the baseline embeddings. Specifically, the p values of SimTeG on OGBN-Arxiv and OGBL-Citation2 are close or larger than 0.05. This further supports our key findings: in cooperation with advanced text embeddings, one can bypass the necessity of using complex GNN models. We have also added the p-value results to the paper (Section A1.2).
>
> | Dataset       | Backbone              | SimTEG | Baseline (GIANT/OGB) | P-Value         | P < 0.05 |
> |---------------|---------------------------------|------------------------|----------------------|-----------------|----------|
> | OGBN-Arxiv    | SOTA GNN   | 77.04               | 75.93             | 7.77e-14        | True     |
> | OGBN-Arxiv    | GraphSAGE  | 76.84               | 73.70             | 4.11e-17        | True     |
> | OGBN-Products | SOTA GNN   | 85.40               | 86.12             | 4.15e-06        | True     |
> | OGBN-Products | GraphSAGE  | 84.59               | 82.84             | 9.01e-10        | True     |
> | OGBL-Citation | SOTA GNN   | 91.42               | 90.92             | 0.0023          | True     |
> | OGBL-Citation | GraphSAGE  | 91.62               | 85.13             | 5.01e-12        | True     |
>
> | Dataset  | Method   | GraphSAGE | SOTA GNN | P-Value  | P < 0.05 |
> |---------------|--------|--------|------|----------|----------|
> | OGBN-Arxiv    | SimTEG   | 76.84 | 77.04  | 0.0427   | True     |
> | OGBN-Arxiv    | Baseline  | 73.70 | 75.93   | 7.79e-22 | True     |
> | OGBN-Products | SimTEG    | 84.59    | 85.40  | 8.74e-06 | True     |
> | OGBN-Products | Baseline  | 82.84   | 85.40   | 7.87e-17 | True     |
> | OGBL-Citation | SimTEG    | 91.62  | 91.42   | 0.1380   | False    |
> | OGBL-Citation | Baseline  | 85.13    | 90.92    | 2.98e-15 | True     |
>
> It is worth noting that as the results of GLEM are reported by the original paper and we do not have the results for each individual experiment, we are not able to compute the corresponding p values. We do acknowledge that there is a subtle difference between SimTeG and GLEM and GLEM outperforms SimTeG on OGBN-Products. This phenomenon is discussed in our observation 4 in Section 5.1 of the paper
>
> ## Comment #2: Could you report the performance of GraphFormers?
>
> Thanks for the suggestions. GraphFormer is closely related to our research line and we report the performance of GraphFormers as follows. The results of GraphFormer on OGBN-Arxiv and OGBN-Products are directly borrowed from the GLEM paper [1] since the datasets and split are exactly the same. We run GraphFormer on ogbl-citation2 for 10 times and report the mean with std. For the hyperparameter setting, we use the default parameters, and the batch size is set to 100 to make it consistent with the reported results in GLEM. As shown in the table, SimTeG performs consistently better than GraphFormer. It is possibly because 1) the GNN-nested architecture of GraphFormer solely allows 1-hop message passing, which limits the express ability of GNN models; 2) GraphFormer’s implementation modifies the architecture code of Bert and cannot be easily extended to other SOTA embedding models nowadays.
>
> We have added the GraphFormer results to Table A8 in the paper.
>
> | method | ogbn-arxiv | ogbn-products | ogbl-citation (MRR) |
> |---|---|---|---|
> |GraphFormer | 72.81 ± 0.20 | 74.72 ± 0.16 | 82.78 ± 0.24 |
> | SimTeG + GraphSAGE | 76.84 ± 0.34 | 84.59 ± 0.44 | 85.13 ± 0.73 |
> | SimTeG + SOTA GNN | 77.04 ± 0.13 | 85.40 ± 0.28 | 86.66 ± 1.21 |
>
> ## Comment #3: Could you explore other PEFT strategies to check their effect on the overfitting problem?
>
> Many thanks for the suggestion. It would be beneficial to understand other PEFT strategies to check the effects. Due to the time limitation, we conducted the experiments for PEFT with IA3 [2] on OGBN-Arxiv. The results are as follows. As the case study showed, different PEFT may result in similar effects since all of them restrict the learnable parameters of LMs.
>
> |   |  LoRA | IA3 |
> |---|---|---|
> | LM stage | 74.06 ± 0.13 | 74.23 ± 0.18  |
> | GNN stage | 76.84 ± 0.34 | 76.54 ± 0.24  |
>
> [1] Zhao, Jianan, et al. "Learning on large-scale text-attributed graphs via variational inference." arXiv preprint arXiv:2210.14709 (2022).
> [2] Liu, Haokun, et al. "Few-shot parameter-efficient fine-tuning is better and cheaper than in-context learning." Advances in Neural Information Processing Systems 35 (2022): 1950-1965.

---

### Author Response · Authors · 2023-11-18
**General Response to the Reviewers**

# General Response to the Reviewers
We sincerely appreciate the time and valuable efforts of the reviewers. We have revised the manuscript accordingly and tried to address all reviewer’s comments and suggestions, both in the manuscript and in the responses. The revision to the manuscript is marked with *blue* text with better readability.  Below we present a general response on the motivation and contribution of this paper, and we respond to each reviewer separately on other concerns.

## Motivation
Regarding the common comments from reviewers about the technical novelty and motivation of this paper, we would like to clarify them here.

We would like to emphasize that *the primary objective of this paper is not proposing a brand new technique to achieve SOTA results on the leaderboard.* Instead, we aim to study a straightforward yet underexplored question: Considering the advancements in Graph Neural Networks (GNNs) are predominantly centered around refining GNN architectures while relying on outdated text embeddings like skip-gram [1,2,3], what outcomes might arise from solely enhancing the text embeddings used in benchmarks? Though it is anticipated that using advanced text embeddings would boost the performance of GNNs, significant questions remain open: how much improvement could we achieve, especially with a simple GNN architecture? *Are advanced text embeddings good alternatives to complicated GNN architecture?*.

These questions cannot be directly answered by recent papers that integrate LM and GNN, including GIANT[4], GLEM [5], and concurrent works [6,7]. However, these are practical and valuable questions, because few GNN architectures have been successfully applied to industry in practice due to their computing efficiency as far as we know (except for Pinsage which is a recommender system based on GraphSAGE for Pinterest). Furthermore, an in-depth examination of these questions could encourage more future researchers to shift their focus towards text embeddings, rather than predominantly concentrating on the architectures of GNNs. This redirection of focus might unveil new insights and advancements in the field.To this end, we develop this straightforward framework, which, as our claim in the abstract, does not innovate in frameworks, models, and tasks.

## Contributions
From our point of view, the main contribution of our works are three-folds:

1. We show that  the **through incorporation of advanced text features, we could bypass the necessity of using complex GNN architectures**. This finding is backed with our experiment results as shown in Table 1 and 2, where a simple two-large GraphSAGE could achieve on-par SOTA performance. This finding indicates the potential of applying a simple and extensible GNN model in more industrial and textual settings.

2. We empirically demonstrate that PEFT is effective for alleviating the overfitting problem when finetuning LMs and further boosting the performance of GNNs on downstream tasks.

3. We systematically examine the impact of various LM variants on enhancing text embeddings and GNN performance. Our findings indicate that LMs pretrained or fine-tuned on text embedding tasks yield more significant improvements in GNN performance, while model size does not emerge as a critical factor.

To make our contribution clearer, we have moderately adjusted our introduction in our manuscripts.

[1] Hu, Weihua, et al. "Open graph benchmark: Datasets for machine learning on graphs." Advances in neural information processing systems 33 (2020): 22118-22133.
[2] Zeng, Hanqing, et al. "Graphsaint: Graph sampling based inductive learning method." arXiv preprint arXiv:1907.04931(2019).
[3] Shchur, Oleksandr, et al. "Pitfalls of graph neural network evaluation." arXiv preprint arXiv:1811.05868 (2018).
[4] Chien, Eli, et al. "Node feature extraction by self-supervised multi-scale neighborhood prediction." arXiv preprint arXiv:2111.00064 (2021).
[5] Zhao, Jianan, et al. "Learning on large-scale text-attributed graphs via variational inference." arXiv preprint arXiv:2210.14709 (2022).
[6] He, Xiaoxin, et al. "Explanations as Features: LLM-Based Features for Text-Attributed Graphs." arXiv preprint arXiv:2305.19523 (2023).
[7] Chen, Zhikai, et al. "Exploring the potential of large language models (llms) in learning on graphs." arXiv preprint arXiv:2307.03393 (2023).

---

### Meta-Review · Area_Chair_2JQ5 · 2023-12-05

**Metareview:**

This paper presents SimTeG, a simple but effective approach aimed at improving textual graph learning with language models. Specifically, the authors first perform Parameter-Efficient Fine-Tuning (PEFT) of language models (LM) on downstream tasks, and then utilise the LM output as input features to GNNs for the same tasks.

While I appreciate the authors’ detailed elaboration on the contributions of their work, and recognise that the findings are interesting, the paper is very similar to previous work and the novelty of the approach is very limited; therefore i would have to agree with reviewers wDRw and PJGj, in addition to the concerns about motivation and evaluation (ie, that model comparisons are not fair and that the proposed approach might have an unfair advantage). I’d think that this paper would be better suited as a workshop paper.

**Justification For Why Not Higher Score:**

The paper is very similar to previous work and the novelty of the approach is very limited. Additionally, there are concerns about evaluation and whether the proposed approach has an unfair advantage.

**Justification For Why Not Lower Score:**

NA

---

### Decision · Program_Chairs · 2024-01-16

Reject